# Metabolic Signature Differentiated Diabetes Mellitus from Lipid Disorder in Elderly Taiwanese

**DOI:** 10.3390/jcm8010013

**Published:** 2018-12-21

**Authors:** Chi-Jen Lo, Hsiang-Yu Tang, Cheng-Yu Huang, Chih-Ming Lin, Hung-Yao Ho, Ming-Shi Shiao, Mei-Ling Cheng

**Affiliations:** 1Metabolomics Core Laboratory, Healthy Aging Research Center, Chang Gung University, Taoyuan City 333, Taiwan; chijenlo@gmail.com (C.-J.L.); tangshyu@gmail.com (H.-Y.T.); chenyu7015@gmail.com (C.-Y.H.); hoh01@mail.cgu.edu.tw (H.-Y.H.); msshiao@gap.cgu.edu.tw (M.-S.S.); 2Division of Internal Medicine, Chang Gung Memorial Hospital, Taipei 105, Taiwan; lin789@adm.cgmh.org.tw; 3Department of Health Management, Chang Gung Health and Culture Village, Taoyuan City 333, Taiwan; 4College of Medicine, Chang Gung University, Taoyuan City 333, Taiwan; 5Graduate Institute of Biomedical Sciences, College of Medicine, Chang Gung University, Taoyuan City 33302, Taiwan; 6Clinical Metabolomics Core Laboratory, Chang Gung Memorial Hospital, Taoyuan City 33302, Taiwan; 7Department of Biomedical Sciences, College of Medicine, Chang Gung University, Taoyuan City 33302, Taiwan

**Keywords:** metabolomics, lipid disorder, diabetes mellitus, elderly

## Abstract

Aging is a complex progression of biological processes and is the causal contributor to the development of diabetes mellitus (DM). DM is the most common degenerative disease and is the fifth leading cause of death in Taiwan, where the trend of DM mortality has been steadily increasing. Metabolomics, important branch of systems biology, has been mainly utilized to understand endogenous metabolites in biological systems and their dynamic changes as they relate to endogenous and exogenous factors. The purpose of this study was to elucidate the metabolomic profiles in elderly people and its relation to lipid disorder (LD). We collected 486 elderly individuals aged ≥65 years and performed untargeted and targeted metabolite analysis using nuclear magnetic resonance (NMR) spectroscopy and liquid chromatography—mass spectrometry (LC/MS). Several metabolites, including branched-chain amino acids, alanine, glutamate and alpha-aminoadipic acid were elevated in LD compared to the control group. Based on multivariate analysis, four metabolites were selected in the best model to predict DM progression: phosphatidylcholine acyl-alkyl (PC ae) C34:3, PC ae C44:3, SM C24:1 and PCae C36:3. The combined area under the curve (AUC) of those metabolites (0.82) was better for DM classification than individual values. This study found that targeted metabolic signatures not only distinguish the LD within the control group but also differentiated DM from LD in elderly Taiwanese. These metabolites could indicate the nutritional status and act as potential metabolic biomarkers for the elderly in Taiwan.

## 1. Introduction

Aging is a complex biological process driving changes in systemic metabolism. Early molecular studies related to ageing have focused on individual genes that significantly affect age-related diseases and longevity [1]. However, the variation in gene expression during in healthy ageing may differ among populations. Therefore, determining the specific impacts of slight environmental changes on age and age-related diseases has proven difficult [2]. 

Metabolomics is a powerful tool for studying metabolic processes, identifying crucial biomarkers responsible for metabolic characteristics and revealing metabolic mechanisms [3]. An important branch of systems biology, metabolomics can measure a large number of low-molecular-weight metabolites, which provides a broad range of information on integrated cellular response. This can be used to understand endogenous metabolites in biological systems and their dynamic changes in relation to internal and external environmental factors [4]. Currently, application of high-throughput metabolic profiling has led to advances in disease diagnostics, biomarker discovery, toxicology, plant science and drug development [5]. In fact, specific metabolomic profiling has been widely used in the study of a variety of degenerative diseases, including cardiovascular and coronary artery disease [6,7,8], diabetes [9], dementia [10] and cancers [11,12]. 

It is estimated that Taiwan will become a hyper-aged society in 2025, where at least 20% of the existing population will be over 65 years of age. Aging is accompanied by several disabilities and chronic diseases such as diabetes and atherosclerosis. Diabetes is the fifth leading cause of death in elderly Taiwanese. Diabetes mellitus (DM) can cause systemic organ damage leading to cardiovascular diseases and nephropathy, which particularly increase in incidence and prevalence in the elderly [13]. Due to the high association of diabetes with metabolic syndrome (MetS), the incidence of MetS increases with age and should be considered an important issue in the elderly individuals. Previous studies have reported high prevalence of MetS-related risk factors in elderly Taiwanese [14,15]. As such, the identification and treatment of patients with MetS would be an important approach to reduce morbidity and impairment in the elderly. MetS is defined as a combination of central obesity, dysglycemia, dyslipidemia and elevated blood pressure [4]. Dyslipidemia is a major contributor to the pathogenesis of metabolic disease and increased incidence of diabetes. Dyslipidemia is characterized by increased levels of triglycerides (TGs) associated with decreased levels of HDL-cholesterol (HDL-C). Despite the decreased levels of LDL-cholesterol, dyslipidemia has been implicated in the pathogenesis of cardiovascular events in adults and children [16,17,18]. Metabolite profiling using nuclear magnetic resonance (NMR) spectroscopy or liquid chromatography—mass spectrometry (LC/MS) can be performed in an untargeted or targeted manner [19]. In this study, we used targeted and untargeted metabolomics to assess the metabolic signature of abnormal TG and HDL-C levels in the elderly, which together are referred to as lipid disorder (LD).

The association between diabetes mellitus (DM) and insulin resistance (IR) has been investigated using metabolic profiling [20,21,22,23,24,25,26,27]. To date, published findings suggest that branched-chain amino acids (BCAAs) and aromatic amino acids are related to the risk of prediabetes and type-2 diabetes [20,21,27]. Several studies have also revealed that lipid subclasses such as phospholipids, sphingomyelins, acylcarnitines and triglycerides have all been associated with insulin resistance and diabetes [23,24,25]. Metabolomics analyses have shown that sugar metabolites are associated with increased risk of DM [22,24]. However, metabolomic studies in elderly populations are limited and seldom investigate the relationship between dyslipidemia and the metabolic profile. Rapid growth of the elderly population in Taiwan demands substantial amounts of healthcare resources, which will increase relative to the proportion of aged Taiwanese. To get better insight into the complex age-associated changes in metabolites and metabolic regulation during LD, we studied a cohort of people aged ≥65 years. Reflecting the metabolic status of the whole system will help predict subjects at increased risk of diseases and provide means for health management in the elderly.

The data combined untargeted and targeted metabolomics methods, using high-throughput screening NMR data acquisition and high-resolution multiple reaction monitoring (MRM) transitions for targeting multiple metabolites, that can find the elderly subjects’ wider profile of metabolic state and their metabolic track from normal to LD and LD with DM. BCAAs, glutamate, alanine and alpha-aminoadipic acid levels increased and those of phospholipids decreased in plasma from the LD group. Multivariate analysis showed that four metabolites were selected in the best model to predict DM progression: phosphatidylcholine acyl-alkyl (PC ae) C34:3, PC ae C44:3, SM C24:1 and PCae C36:3. These results suggest that the metabolic signature not only distinguished the LD group from the control group but also differentiated diabetes from LD in the elderly.

## 2. Materials and Methods

### 2.1. Ethics Statement and Study Populations

The study protocol was approved by the Institutional Review Boards of Chang Gung Memorial Hospital. Written informed consent was obtained from all recruited participants. Following at least one year of residency in the Chang Gung health and culture village in Taiwan, patient’s blood samples were collected during their scheduled yearly physical and health examinations. The participants of this study took questionnaire about their health and voluntarily disclosed their medication information when they were first enrolled. After conclusion of questionnaire survey, the medication information of participants was largely incomplete and not amenable to analysis. Individuals with a Body Mass Index (BMI) >35, BMI <15, blood glucose level >300 mg/dL or for whom clinical data were lost were excluded from the study. This study enrolled a total of 486 subjects aged ≥65 years (Figure 1). 

MetS was defined according to criteria from the National Cholesterol Education Program’s Adult Treatment Panel III (NCEP ATP III) [28]. Plasma glucose levels were modified in accordance with the American Diabetes Association that updated the definition of impaired fasting glucose (IFG) [29] and adapted cut points for abdominal obesity for Asian populations [30]. Subjects were identified as MetS by having at least three of the following five risk factors: (1) Waist circumference (WC) ≥90 cm for men or ≥80 cm for women; (2) systolic blood pressure (SBP) ≥130 mmHg or diastolic blood pressure (DBP) ≥85 mm Hg; (3) fasting plasma glucose level ≥100 mg/dL; (4) plasma HDL-C <40 mg/dL for men or <50 mg/dL for women; and (5) plasma TG level ≥150 mg/dL. Participants with abnormal levels of both TG and HDL-C in plasma were referred to as the LD group, whereas, participates without any risk factors were referred to as the control group (Table 1).

### 2.2. Nuclear Magnetic Resonance (NMR) Analysis of Plasma

Plasma samples from 423 participants in the aged cohort were collected into EDTA tubes (BD Vacutainer, Franklin Lakes, NJ, USA) and stored at −80 °C until NMR analysis. Plasma samples (350 μL) were mixed with 350 μL of buffer (75 mM Na_2_HPO_4_, 0.08% 3-(Trimethylsilyl)propionic-2,2,3,3-d4 acid (TSP), 2 mM NaN_3_ and 20% D_2_O) and 600 μL of the supernatant was transferred to 5 mm NMR tubes for analysis. The daily plasma quality control (QC) sample was an equal mixture of all samples collected on a given day.

^1^H NMR spectra were acquired on a Bruker Avance III HD 600 MHz NMR spectrometer (Bruker Biospin GmbH, Rheinstetten, Germany) at a temperature of 310K using a 5 mm inverse triple resonance CryoProbe (^1^H/^13^C/^15^N) with cold preamplifier for ^1^H and ^13^C with a z-axis gradient (Bruker Biospin GmbH, Rheinstetten, Germany). The spectra were acquired by Carr-Purcell-Meiboom-Gill (CPMG) spin-echo pulse sequence and a total T_2_ relaxation time of 80 ms was used to attenuate broad signals. The ^1^H NMR spectrum was collected using a spectral width of 12,019.23 Hz, a relaxation delay of 4.0 s and an acquisition time of 2.7 s. Free induction decay (FID) was acquired into data points with a dimension of 72 K and the FID acquisitions were accumulated 32 times to increase the signal-to-noise ratio. FIDs were weighted by an exponential function with a 0.3 Hz line-broadening factor prior to Fourier transformation and the dimension of the process data points was 128 K. Finally, the first-order phase was set to 0 and the automatically phased was using only zero-order phase correction. All NMR spectra were phased and baseline-corrected using the Topspin software (version 3.2.2, Bruker Biospin GmbH, Rheinstetten, Germany) [31] and then referenced to the doublet of 1H-α-glucose at a chemical shift of 5.23 ppm. After processing, the spectra should satisfy the criteria that the line width at half height of lactate resonance at 1.32 ppm was <1.15 Hz (Appendix A). The spectra of plasma QC for each sample were acquired at the beginning and the end of each day, in order to evaluate the variability between the first and last samples of the day (Appendix A). Each ^1^H NMR spectrum was segmented into equal widths (0.01 ppm), corresponding to regions 0.5–9.5 ppm, providing 900 features in each NMR spectrum. The spectral data were normalized to TSP by the AMIX software (Bruker Biospin GmbH, Rheinstetten, Germany). The water signal region (4.5–5.0 ppm) was removed before statistical analysis. The resonant frequencies of each metabolite were obtained from the library of the Chenomx NMR Suite 7.1 (Chenomx, Edmonton, AB, Canada). The data required from researchers are available from corresponding author.

### 2.3. Determining Concentrations of Plasma Metabolites with Ultra-High-Performance Liquid Chromatography-Tandem Mass Spectrometry (UPLC/MSMS)

A total of 82 plasma samples were analyzed using a commercially available kit (AbsoluteIDQ p180, BIOCRATES Life Sciences AG, Innsbruck, Austria). The kit was used to quantify 184 metabolites including amino acids, biogenic amines, glycerophospholipids, sphingolipids, acyl carnitines and hexose. The samples were processed according to the manufacturer’s instructions as previously described [8]. Briefly, each 10 μL of plasma was prepared according to the manufacturer’s instructions. The levels of biogenic amines and amino acids were determined by UPLC/MSMS and standard chromatogram (Appendix A) and other lipid species were quantified by flow injection analysis coupled with tandem mass spectrometry (FIA-MSMS). The plasma samples were standardized by spiking in isotopically-labeled standards. The analysis was performed under positive electrospray ionization mode with scheduled multiple reaction monitoring (Waters Crop., Milford, CT, USA). Chromatographic separation was performed using an Acquity BEH C8 column (75 mm × 2.1 mm, particle size of 1.7 μm; Waters Crop., Milford, CT, USA) at 50 °C using a gradient mixture of water with 0.2% formic acid and acetonitrile with 0.2% formic acid at a flow rate of 0.9 mL/min using a linear gradient: 0–0.38 min: 0% B; 0.38–3 min: 0–15% B; 3–5.4 min: 15–70% B; 5.4–5.93 min: 100% B; 5.93–6.6 min: 0% B for re-equilibration. The mass parameters were as follows: Capillary of 3.2 kV; desolvation gas flow of 1200 L/h; desolvation temperature of 650 °C; source temperature of 150 °C; and cone voltage of 10 V, respectively. For FIA analysis, a flow rate of 0.03 mL/min was used with commercial solvent. The mass parameters were as follows: Capillary of 3.9 kV; desolvation gas flow of 650 L/h; desolvation temperature of 350 °C; source temperature of 150 °C; and cone voltage of 20 V, respectively. The measurements were made in a 96-well format. One blank was used (without internal standards) to calculate the background level and to check the system for contamination. Three zero samples (PBS) were used to calculate the limit of detection, seven calibration standards, three levels of quality control samples (low, medium, high), all of which were integrated into the plate. The MetIQ software 6.0.0-DB104-Carbon-2743 (Biocrates Life Science AG, Innsbruck, Austria) automatically checked whether the measured values of blank, standards and QC samples were within the ranges (Appendix A). Metabolite concentrations were calculated and expressed as μM and values below the limit of detection (<LOD) or below the lower limit of quantification (<LLOQ) were excluded. The remaining 154 metabolites were finally analyzed. Finally, the raw data could be available by connecting the corresponding author.

### 2.4. Statistical Analysis

The student’s *t*-test was performed to compare laboratory data between the control and LD groups. The visualization model included principal component analysis (PCA) and orthogonal partial least squares discriminant analysis (OPLS-DA) model, which were performed using the SIMCA-P software (version 13.0, Umetrics AB, Umea, Sweden). A student’s *t*-test or one way analysis of variance (ANOVA) with Tukey’s post hoc test was applied for metabolite analysis for NMR and LC/MS spectrometry. Multivariable generalized linear modeling analysis was used to adjust for age and comorbidities, including hypertension, coronary artery disease (CAD), stroke, chronic kidney disease (CKD). Model significance was presented as adjusted *p*-value. The analyses were performed using the SAS 9.4 software (SAS Institute, Cary, NC, USA). For the large number of statistical comparisons, the calculation of the false discovery rate (FDR) was applied to the *p*-values to correct for multiple testing (*q*-value). To explore prognostic metabolites for DM progression, a multiple logistic regression model with backward elimination was applied to the data sets. To identify independent predictor of DM group, linear logistic regression analysis was performed. Receiver operating characteristic (ROC) curve was constructed and the area under the ROC was used to measure the predictive accuracy and compared between control and DM groups. 

## 3. Results

A total of 486 participants were included in this analysis and grouped according to five risk factors for MetS. Within the LD group (abnormal levels of TG and HDL-C), 47 subjects (95.9%) exhibited factors associated with MetS (Table 1). Moreover, elderly patients were more likely to have DM association in the LD group (49%), in comparison to the control (5.3%) and other group (23.9%). Relative to the control group, the LD group had significantly higher weight, BMI, WC, SBP, DBP, plasma glucose, Hb-A1c, TG, plasma albumin, total plasma protein, blood urine nitrogen (BUN), plasma creatinine and plasma uric acid (UA), as well as significantly lower plasma HDL-C and total bilirubin values (Table 1).

To improve the prediction of DM incidents in elderly people, we characterized the metabolic profile of the participants using NMR. Multivariate analysis methods such as PCA and OPLS-DA score plots, represented progressive change from control, other and LD groups, respectively (Figure 2A,B). The data showed that these metabolites could clearly discriminate between the control and LD groups in OPLS-DA (Figure 2B). The different metabolites between the control and LD groups were identified and were represented in Figure 2C. These metabolites include amino acids and varied lipids (Appendix A). Amino acids, such as alanine, histidine, isoleucine, leucine, valine, phenylalanine and tyrosine were significantly increased in the LD group compared to control group.

The high-throughput screening of untargeted metabolites with NMR revealed that amino acids and varied lipids were effective for discriminating the LD and the control group. However, a disadvantage of our untargeted analysis using NMR was that different types of lipid cannot be discriminated from those functional groups and could fail at detecting some low-level metabolites by NMR analysis. Since most metabolites identified by our untargeted NMR approach belong to categories of lipids and amino acids in the AbsoluteIDQ^®^ p180 Kit, we subsequently used this kit to quantify the metabolites and further define their relationship. Samples from a total of 82 subjects were included in LC/MSMS analysis. Due to the fact that nearly 50% of the LD group subjects had DM, we divided the LD subjects into two groups; with or without DM. We used this approach to clarify which phospholipids and amino acids could be used to discriminate control from LD groups. The PCA and OPLS-DA score plots demonstrated a considerable separation between the control, LD and LD with DM groups, however the control with DM group could not be separated from the other groups (Figure 3A,B). This data revealed that the strongest differences were between the control and LD groups and comprised of 154 metabolites. After adjusting for age, hypertension, CAD, stroke and CKD, 47 of the 154 metabolites were significantly different between the control and LD groups, with a false discovery rate (FDR) to correct the *p*-value (*q* < 0.05) (Figure 4 and Appendix A). Amino acids such as glutamate, alanine, alpha-aminoadipic acid, valine, leucine and phenylalanine were significantly increased in the LD compared to the control group. Whereas, phospholipids such as PC ae C34:3, PC ae C44:6, PC ae C42:4, PC ae C32:2, PC ae C44:3, PC ae C44:4, PC ae C34:2, PC ae C42:3, PC ae C36:3, PC ae C38:2, PC ae C44:5, PC ae C32:1, PC ae C42:5, PC ae C40:4, PC ae C40:3, PC ae C38:1, PC ae C42:2, PC ae C40:5, PC ae C36:2, PC ae C40:2, Sphingomyelin (SM) C16:0, SM C24:1, SM C26:1, PC diacyl (aa) C42:0, PC aa C42:2, PC aa C42:1, PC aa C40:1, PC aa C40:2 and PC aa C40:3 were markedly lower in the LD group. And we showed top 10 significantly different metabolites in Figure 5.

To find out which metabolites were related to the progression from the LD group using the 47 significantly different metabolites identified between the LD and control groups (Appendix A), we used a multiple logistic regression model with backward elimination. Four metabolites were selected in the best model: PC ae C34:3, PC ae C44:3, SM C24:1 and PC ae C36:3. Table 2 shows the odds ratio and 95% confidence interval (CI) and ROC curves for four metabolites with higher sensitivity and specificity than four individual metabolites (Figure 3C). This data emphasizes that combinations of features can give more information than features considered singly.

## 4. Discussion

In this study, we combined untargeted and targeted metabolomics methods by collecting data for high-throughput screening NMR data acquisition and high-resolution MRM transitions for targeted multiple metabolites, to obtain a wider profile of elderly subjects’ metabolic state and their metabolic track from the normal to LD and LD with DM groups. Multivariate analysis confirmed that the LD group presented with BCAAs, aromatic amino acids and lipidome alteration (Figure 3D). 

Metabolites, such as amino acids, are well established biomarkers for DM and insulin resistance [32]; in particular, the BCAA-related metabolites, including valine and leucine, that were highly associated with LD in the elderly, have been related to insulin resistance [18,33,34]. Furthermore, a complete analysis of acylcarnitine species by MS/MS analysis revealed a clear increase in C3 and C5 acylcarnitines (short-chain acylcarnitines) in the LD and LD + DM groups compared to the control groups. Branched-chain aminotransferase present in muscle tissue can metabolize BCAAs to branched-chain keto acids and those of keto acids, which are then oxidized to acylcarnitines (C3 and C5) by branched-chain keto acid dehydrogenase [35]. These results suggest that C3 and C5 acylcarnitines are direct products of BCAA catabolism. In our results, the levels of energy related-metabolites, including glucose, glutamate, alanine and pyruvate were significantly higher in the LD compared to the control group. A high rate of BCAA catabolism results in the release of ammonia to synthesize glutamate from alpha-ketoglutarate [36]. This accumulation of glutamate may increase transamination of pyruvate to alanine [37]. Increases in the levels of alanine, a highly gluconeogenic amino acid responsible for glucagon to regulate gluconeogenesis, could contribute to the development of glucose intolerance in LD [38]. Indeed, circulating pyruvate and alanine levels were clearly elevated in the LD subjects. In the beta cell, glutamate is synthesized and transported into insulin-secreting granules [39] and released with insulin to increase the insulin and glutamate levels in circulation. 

Dyslipidemia extends far beyond cholesterol and TGs. Phospholipid and sphingolipid metabolism and how metabolic disturbances were found in obesity, prediabetes and DM, are currently known and discussed [40]. LD was usually associated with higher levels of TGs and lower levels of HDL-C. In our study, elderly participants with LD presented lower levels of long-chain fatty acids and polyunsaturated phosphatidylcholines. Several studies have reported that prediabetes and type-2 DM are associated with a decrease in phosphatidylcholine [23,24,41]. Negative associations with vinyl ether-linked phosphatidylcholines (plasmalogens), LD and DM were also observed [23,24]. Levels of plasmalogens and SMs were significantly lower in the LD compared to control group and the multiple logistic regression analysis showed the model of PC ae C34:3, PC ae C44:3, SM C24:1, PC ae C36:3 was better than one of them for DM classification (the area under the curve of those metabolites is 0.82). Levels of these four metabolites were lower as DM progressed from LD and LD + DM. The change in metabolites may play vital roles in disease progression, such as that from normal to LD and from LD to DM. Plasmalogens in plasma are primarily synthesized in the liver and secreted into circulating blood as lipoprotein components. The plasmalogens are particularly susceptible to oxidative stress and serve as endogenous antioxidants and mediators of membrane structure and dynamics and protect membranes and lipoproteins from oxidation [42,43]. Reduced plasmalogen abundance in the LD group might reflect an increased oxidative imbalance, probably due to a systemic inflammatory response. A low plasmalogen level has been reported to be responsible for the oxidative status during aging [44]. Metabolic diseases such as prediabetes, type-2 DM [41] and cardiovascular disease [45] present with elevated oxidative stress and were negatively associated with plasmalogen levels. Plasmalogen levels in plasma are known to be positively correlated with serum levels of HDL-C [46] and more strongly associate with multiple risk parameters other than HDL-C [47]. This suggests that plasmalogens might play the role of antioxidants and control the efflux of cholesterol [48].

Sphingomyelins in populations decreased in the DM group [24,49]. This is consistent with the results of our study, which demonstrated that sphingomyelins were negatively associated with diabetes progression in the LD group. 

The prevalence of MetS has been reported to be associated with age and sex [50]. In Taiwan, the prevalence of MetS in men between 40 and 64 years old has been shown to increase from 29.7% to 45.5%, respectively, however the incidence dropped from 45.5% to 36.8% in men who were over 65 years old [50]. In Taiwanese women, the prevalence of MetS has been found to be continuously increasing. This finding was confirmed in another large cohort study, which enrolled 18,907 elderly Taiwanese [14]. Since the metabolic profiling of aged groups may be different to middle-age group, the changed in metabolites may be different to the profile seen in the general population such as middle-age individuals or children. Although several cohorts have given prospective and retrospective studies and reveal significant differences in metabolites in MetS and DM, they were unable to present the metabolic status in the elderly group. Our study is the first to reveal the metabolic profiling of the super-elderly participants (mean age of over 80) in a care institution. In this population, the incidence of MetS (~29%) was relatively lower than in the elderly Taiwanese, who are apparently healthy. It is also the first study to shed light on the metabolic signature of LDs with diabetes in super-elderly Taiwanese.

## 5. Conclusions

The strength of our study is the prospective presentation in elderly participants living in the same village, receiving the same medical care and having the similar lifestyles. We have minimized the environmental factors as much as possible. However, there are some limitations. The sample size is small in this cohort, since this study was a preliminary study; however, the four-year follow-up monitoring is continuing and the metabolic signatures of LD and LD with DM in this study will be continuously investigated in the future. We are also aware of the risk of overfitting the classifier model to our limited set of subjects, despite attempts to minimize these effects with the proper statistical methods used. Furthermore, these analyses could not take into account all of the possible confounding factors, such as clinical laboratory data and medication. The participants of this study took questionnaire about their medication information. After conclusion of this survey, the medication information of participants was largely incomplete and not amenable to analysis. However, this study provides enormous information about the elderly participants and we believe that it promotes healthy aging research. More importantly, metabolites can indicate nutritional status as well as being potential metabolic biomarkers for elderly individuals. Therefore, we believe that the early identification of high risk groups of LD could provide appropriate intervention and improve the health within communities in order to reduce long-term medical care in Taiwan. 

## Figures and Tables

**Figure 1 jcm-08-00013-f001:**
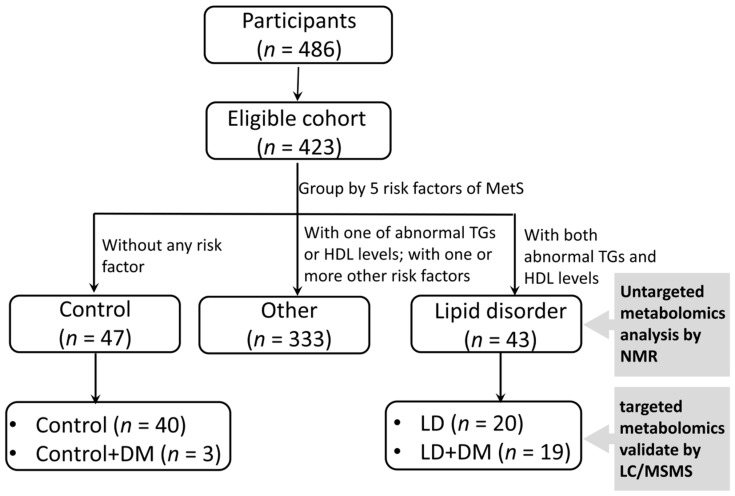
Flow diagram of study selection for metabolite analysis. Plasma samples from all participants were collected during their health examination and analyzed by nuclear magnetic resonance (NMR) spectroscopy and liquid chromatography—mass spectrometry (LC/MS). Initially, for untargeted metabolomic analysis, the plasma from 423 participants was analyzed. The targeted metabolite analysis was performed for 82 participants, including 43 normal controls and 39 lipid disorder (LD) participants. Abnormal values of HDL-C and TG define LD. Participants whose blood glucose level was higher than 300 mg/dL or whose BMI >35 or <15 were excluded from the normal control and LD groups.

**Figure 2 jcm-08-00013-f002:**
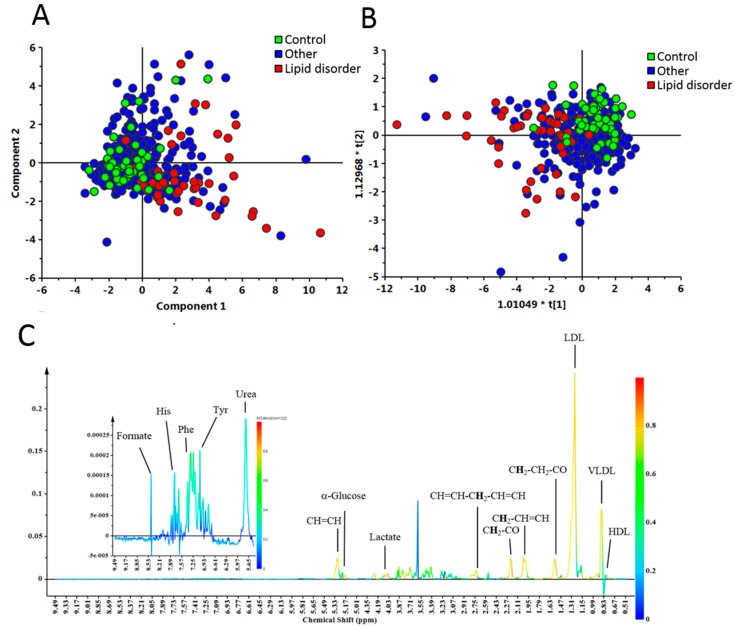
Metabolites were significantly different between the control and lipid disorder (LD) groups. The metabolic profile of the plasma from elderly people was analyzed with nuclear magnetic resonance (NMR) spectroscopy. (**A**) Principal component analysis (PCA) score plot showing progressive change from 47 control (green), 333 other (blue) and 43 LD (red) participants, respectively; (**B**) Orthogonal partial least squares discriminant analysis (OPLS-DA) showing metabolites can clearly discriminate the control and LD groups; (**C**) OPLS-DA coefficient loading plots derived from ^1^H NMR spectra of the control and LD groups. The coefficient values above zero (upper section) represent levels of metabolites that were higher in LD group than control group. The color bar corresponds to the absolute value of the correlation loading in the discrimination model, represent metabolites that were more prevalent in the LD group.

**Figure 3 jcm-08-00013-f003:**
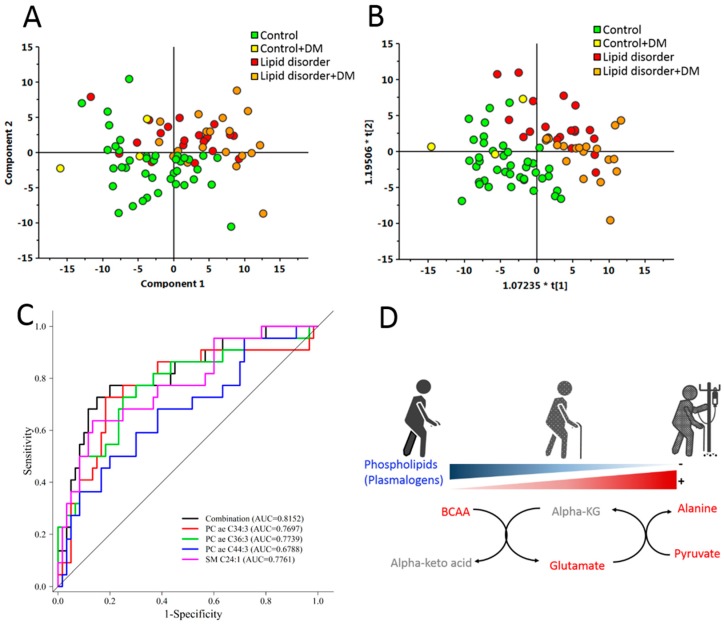
Targeted metabolites were significantly different between control, lipid disorder (LD) and LD with diabetes mellitus (DM) groups. Metabolites in 40 control participants, 3 control with DM participants, 20 LD participants and 19 LD with DM participants were validated with liquid chromatography tandem mass spectrometry. (**A**) Principal component analysis (PCA) and (**B**) orthogonal partial least squares discriminant analysis (OPLS-DA) score plots demonstrated a considerable separation between control, LD and LD with DM groups, however the control with DM group could not be separated from the other groups; (**C**) Receiver operating characteristic (ROC) curve for the predictive model. A combination metabolite model calculated from the logistic regression analysis; (**D**) A schematic diagram illustrating how metabolites change following DM progression from control; metabolites marked in blue refer to those which decreased with DM progression, while metabolites marked in red refer to those which increased with DM progression.

**Figure 4 jcm-08-00013-f004:**
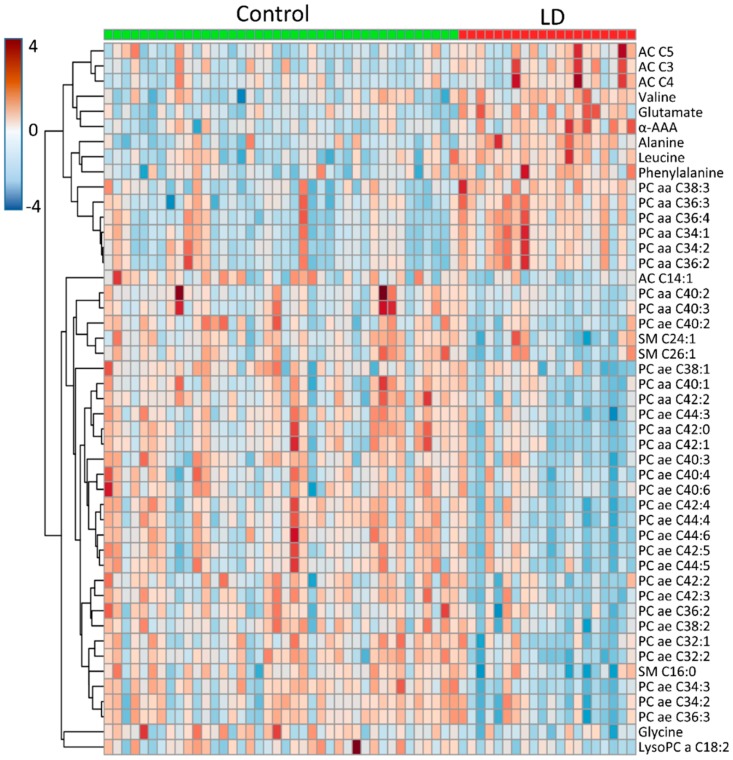
Heat map of 47 metabolites were significantly different between control and lipid disorder (LD) group. After adjusting for age, hypertension, CAD, stroke and CKD, 47 of the 154 metabolites were significantly different between the control and LD groups, with a false discovery rate (FDR) to correct the *p*-value (*q* < 0.05). Each column represents a plasma sample and each row represents the level of a metabolite.

**Figure 5 jcm-08-00013-f005:**
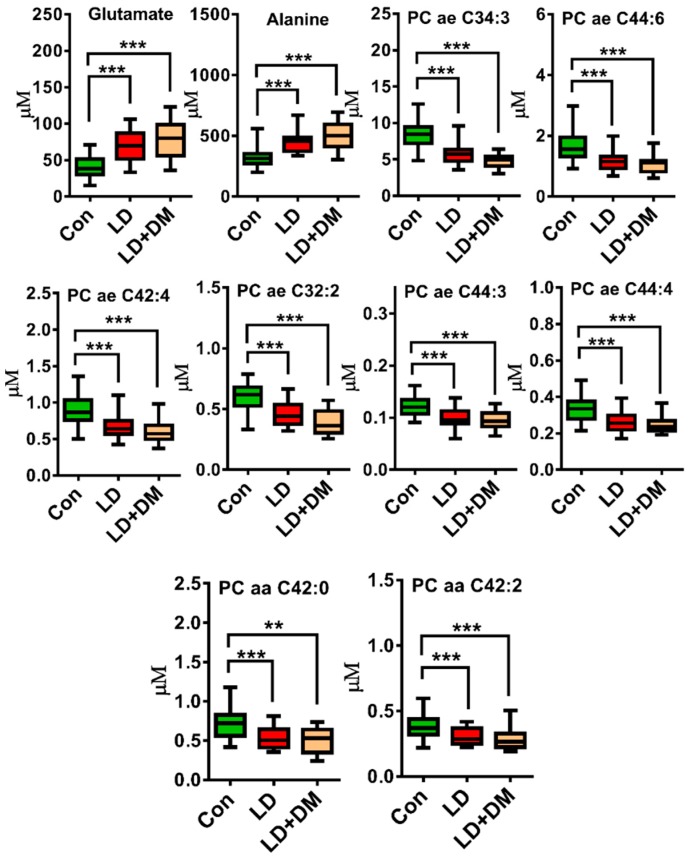
Bar charts of top 10 of 47 metabolites in control (green), lipid disorder (LD) (red) and lipid disorder with DM (LD + DM) (orange). ** *q* < 0.01; *** *q* < 0.001 (*p*-value was adjusted for age, hypertension, CAD, stroke and CKD and was corrected with a false discovery rate (FDR).

**Table 1 jcm-08-00013-t001:** Demographic and laboratory data for an elderly Taiwanese population.

ID	All (*n* = 486)	Control (*n* = 57)	LD (*n* = 49)	Other (*n* = 380)	*p*-Value
Age (years)	81.2 ± 7.1	80.1 ± 7.3	79.8 ± 7.7	81.6 ± 7.0	0.8291
Height (cm)	157.6 ± 8.7	157.1 ± 8.0	154.6 ± 7.7	158.1 ± 8.8	0.1064
Weight (kg)	57.9 ± 10.8	49.8 ± 8.0	62.0 ± 11.8	58.6 ± 10.4	<0.001
BMI (kg/m^2^)	23.2 ± 3.5	20.1 ± 2.4	25.8 ± 3.9	23.3 ± 3.2	<0.001
Waist circumference (cm)	86.8 ± 9.9	75.7 ± 5.1	91.9 ± 11.7	87.8 ± 9.1	<0.001
SBP (mmHg)	134.5 ± 20.8	115.6 ± 8.6	139.4 ± 21.0	136.7 ± 20.6	<0.001
DBP (mmHg)	72.0 ± 10.0	66.1 ± 8.0	72.8 ± 13.7	72.8 ± 9.4	0.0036
Glucose (mg/dL)	99.0 ± 18.1	87.6 ± 6.6	110.2 ± 27.6	99.2 ± 16.8	<0.001
Hb-A1c (%)	6.0 ± 0.7	5.6 ± 0.3	6.4 ± 0.8	6.0 ± 0.7	<0.001
T-cholesterol (mg/dL)	184.7 ± 35.4	195.4 ± 33.2	193.9 ± 40.5	181.9 ± 34.5	0.8275
Triglyceride (mg/dL)	102.9 ± 53.5	80.3 ± 27.9	212.2 ± 57.7	92.2 ± 37.3	<0.001
HDL-C (mg/dL)	54.1 ± 13.7	62.2 ± 12.2	40.4 ± 5.4	54.7 ± 13.4	<0.001
LDL-C (mg/dL)	107.4 ± 29.4	112.8 ± 28.6	109.8 ± 35.8	106.2 ± 28.6	0.6296
Albumin (g/dL)	4.3 ± 0.3	4.3 ± 0.2	4.5 ± 0.2	4.3 ± 0.3	0.0001
Total protein (g/dL)	7.0 ± 0.4	6.9 ± 0.4	7.2 ± 0.4	7.0 ± 0.4	0.0005
AST/GOT (U/L)	27.0 ± 11.4	29.1 ± 18.3	27.2 ± 10.4	26.6 ± 10.2	0.5050
ALT/GPT (U/L)	19.3 ± 13.6	18.1 ± 17.0	21.3 ± 10.2	19.2 ± 13.5	0.2447
ALK-P (U/L)	66.2 ± 21.1	63.2 ± 19.0	67.3 ± 22.2	66.5 ± 21.3	0.3043
Total bilirubin (mg/dL)	0.7 ± 0.3	0.7 ± 0.3	0.6 ± 0.2	0.7 ± 0.3	0.0036
BUN (mg/dL)	17.9 ± 7.4	16.2 ± 4.4	20.9 ± 10.6	17.8 ± 7.2	0.0051
Creatinine (mg/dL)	0.9 ± 0.5	0.8 ± 0.4	1.1 ± 0.7	0.9 ± 0.5	0.0473
Uric acid (mg/dL)	5.7 ± 1.5	5.0 ± 1.3	6.7 ± 2.0	5.7 ± 1.5	<0.001
Hypertension (%)	62.6 (304)	38.6 (22)	77.6 (38)	64.2 (244)	<0.001
Hyperlipidemia (%)	29.8 (145)	15.8 (9)	44.9 (22)	30.0 (114)	0.0012
Metabolic syndrome (%)	28.6 (139)	0 (0)	95.9 (47)	24.2 (92)	<0.001
DM (%)	24.3 (118)	5.3 (3)	49.0 (24)	23.9 (91)	<0.001
CAD (%)	7.6 (37)	0 (0)	10.2 (5)	8.4 (32)	0.0237
Stroke (%)	7.6 (37)	5.3 (3)	10.2 (5)	7.6 (29)	0.3530
CKD (%)	9.9 (48)	3.5 (2)	22.4 (11)	9.2 (35)	0.0049

Data are mean ± SD. Variables were analyzed by independent sample *t*-tests between the control and LD groups. LD, lipid disorder; DM, diabetes mellitus; CAD, coronary artery disease; CKD, chronic kidney disease.

**Table 2 jcm-08-00013-t002:** Multivariate analysis for DM.

Variable	Odds Ratio (95% CI)	*p*-Value
Age	1.016 (0.924, 1.118)	0.7421
Lipid disorder	29.096 (2.635, 321.309)	0.0059
PC ae C34:3	0.488 (0.228, 1.046)	0.0651
PC ae C44:3	<0.001 (<0.001, 0.359)	0.0453
SM C24:1	1.024 (1.001, 1.048)	0.0439
PC ae C36:3	3.112 (1.104, 8.771)	0.0318

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
