# Peer review of "Metabolic Signature Differentiated Diabetes Mellitus from Lipid Disorder in Elderly Taiwanese"

_jcm, 2018, doi:10.3390/jcm8010013_

Reviewer 1 Report

This manuscript by Lo et al describes an interesting and potentially valuable dataset, which measures the blood metabolic profiles via NMR for 423 elderly Taiwanese participants, and targeted LC-MS/MS for a subset of 82 participants. The authors postulate that a metabolic signature can be obtained by comparing 19 participants with diabetes mellitus (DM) and 20 participants without DM, all with high level of blood lipids.

The cohort is typical of an aging population, with prevalent health issues, e.g. over 60% hypertension. The common diseases in this population often overlap, and share risk factors and confounding factors. There is no guarantee that the core design in this study, comparing 19 DM+LD and 20 LD without DM, is not compounded by other disease conditions and risks. The findings in 39 (19+20) participants are expected to be underpowered and possibly only specific to this small group. 

The statistical approach is very inadequate to answer the posed scientific questions.

OPLS-DA is a supervised method, thus its visualization is expected to separate the participants by designated groups. When metabolite features are compared between groups, proper statistical method and correction for multiple comparison (such as False Discovery Rate) should be used. The main piece of data in Figure 3B is not based on proper statistical analysis. The rationale to assign numbers 1-2-3 to the three groups is presumably based on the perception of disease severity, but it’s totally subjective here and without a solid justification. 

Given the complexity of the cohort, I would advise the authors to redo the data analysis using a more mainstream regression method. When lipid levels (this can be total lipids) and DM status are both included in a regression model, one can test the significant contribution to metabolite concentration by each. Prior to constructing the model, the confounders should be tested for their significance and included in the model if appropriate. Obviously, there’s more subjects analyzed by NMR than MS. The regression method can be applied to both data types.

It is also very important to include the medications of the participants, as they have profound impact both to the disease status and on the metabolic profiles.

In reporting results from each of NMR and LC-MS/MS, there should be an overall summary of metabolites measured, and quality metrics. The 650°C for desolvation temperature in positive ESI seems high – please clarify the rationale or add literature reference. 

All analysis needs proper reporting of statistical method, number of subjects and number of features.

Improvement is needed for both the English and the technical terminology:

e.g. “systems biology” not “systematic biology”; metabolites not “differentially expressed”, but “differentially abundant”.

Author Response

Point 1: The cohort is typical of an aging population, with prevalent health issues, e.g. over 60% hypertension. The common diseases in this population often overlap, and share risk factors and confounding factors. There is no guarantee that the core design in this study, comparing 19 DM+LD and 20 LD without DM, is not compounded by other disease conditions and risks. The findings in 39 (19+20) participants are expected to be underpowered and possibly only specific to this small group. The statistical approach is very inadequate to answer the posed scientific questions.

Response 1: we re-analyzed data for 40 control and 20 LD patients, and those for 20 LD patientsand 19 LD+DM patients using a generalized linear mixed (GLM) model, and made multivariate adjustment for other diseases , including age, hypertension coronary artery disease (CAD), stroke, chronic kidney disease (CKD). For making multiple comparisons,  the false discovery rate (FDR) was applied to to make correction for multiple tests.

Point 2: OPLS-DA is a supervised method, thus its visualization is expected to separate the participants by designated groups. When metabolite features are compared between groups, proper statistical method and correction for multiple comparison (such as False Discovery Rate) should be used. The main piece of data in Figure 3B is not based on proper statistical analysis. The rationale to assign numbers 1-2-3 to the three groups is presumably based on the perception of disease severity, but it’s totally subjective here and without a solid justification.

Response 2: We performed unsupervised method (PCA) and supervised method (OPLS-DA) (figure 2A, 2B, 3A, and 3B) to visualize the separation between groups. We applied the false discovery rate (FDR) to make correction for multiple tests (table 2). The statistical methods are described in section entitled2.4. Statistical Analysis”.

Point 4: Given the complexity of the cohort, I would advise the authors to redo the data analysis using a more mainstream regression method. When lipid levels (this can be total lipids) and DM status are both included in a regression model, one can test the significant contribution to metabolite concentration by each. Prior to constructing the model, the confounders should be tested for their significance and included in the model if appropriate. Obviously, there’s more subjects analyzed by NMR than MS. The regression method can be applied to both data types.

Response 4: To study the significance of differences between control and LD group, we used multivariable generalized linear modeling analysis to adjust for age and comorbidities, including hypertension, coronary artery disease (CAD), stroke and chronic kidney disease (CKD), for both NMR and MS data sets. A multiple logistic regression model with backward elimination was applied to analyse data set. Confounders like age, hypertension, CAD, stroke, CKD, and lipid disorder were included in the model. The statistical methods are described in section entitled2.4. Statistical Analysis”.

Point 5: It is also very important to include the medications of the participants, as they have profound impact both to the disease status and on the metabolic profiles.

Response 5: The medication record contains the diagnostic information, clinical data and biochemical data. The Clinical information is now presented in table 1.

Point 6: In reporting results from each of NMR and LC-MS/MS, there should be an overall summary of metabolites measured, and quality metrics. The 650 oC for desolvation temperature in positive ESI seems high – please clarify the rationale or add literature reference. 

Response 6: The overall summary of metabolites measured by LC/MS is shown in supplementary table 3. The quality metrics of NMR data is described in section entitled “2.2. Nuclear Magnetic Resonance (NMR) Analysis of Plasma” and in supplementary table 1. The quality metrics of MS data is described in section entitled “2.3. Determining Concentrations of Plasma Metabolites with Ultra-High-Performance Liquid Chromatography-Tandem Mass Spectrometry (UPLC/MSMS)” and supplementary figure 3.

The parameters for LC/MS analysis of targeted metabolites were set according to those described in user manual of Biocrate AbsolueIDQ p180 Kit with the following modifications. The desolvation temperature stated in the user manual is 600 oC (shown below), however, the ionization under a flow rate of 0.9 ml/min is not efficient. We increased the desolvation temperature to 650 oC, and changed the flow rate of desolvation gas from 1000 L/Hr to 1200 L/Hr. This improved both ionization and sensitivity of our experiment.

Point 7: All analysis needs proper reporting of statistical method, number of subjects and number of features.

Response 7: Proper statistical methods have been used in our study. The statistical methods, number of subjects and number of features are described in sections entitled2.2. Nuclear Magnetic Resonance (NMR) Analysis of Plasmaand 2.4. Statistical Analysis, and in the figures and tables.

Point 8: Improvement is needed for both the English and the technical terminology:

e.g. “systems biology” not “systematic biology”; metabolites not “differentially expressed”, but “differentially abundant”.

Response 8: English is brushed up in the revised manuscript.

Reviewer 2 Report

Dear authors,

I just completed the review of manuscript entitled “Metabolic Signature Differentiated Diabetes Mellitus from Lipid Disorder in Elderly Taiwanese”. The content of the manuscript is important to understand the pathogenesis of metabolic disorder like diabetes. However I have some comments that need to be addressed. Below are the comments:

1)     All the sections in the manuscript should be rewritten. Introduction lacks sufficient review. I would suggest authors to review on metabolites and their implication in disease pathogenesis. The information presented in “Introduction” section is general.

2)     Material and method section lacks sufficient information. The preprocessing of NMR data should be explained in detail with all the parameter settings. The important point I want to highlight is quality control. Were the quality control data run during data acquisition with NMR and LC-MS? All the quality control strategy should be illustrated and the validation data should be presented as supplementary information. The raw data (excel or .csv) should be submitted as supplementary information. Biocrates kit has quality data and should be included in the manuscript as supplementary information. I would also suggest to show overall chromatogram of Biocrates standards (LC-MS, not FIA) as supplementary information.

3)     I would suggest use of more other statistical tool for analysis and infer crucial result. I also would suggest authors to use visual presentation of result than large table. For example, the authors should perform supervised and unsupervised learning to study their data. Principal component analysis would be the best way to see the data clustering. Authors have mentioned about one-way ANOVA but have not highlighted its finding in result at least as visual or tabular form.  The correlation analysis may visually be presented to see the relationship between metabolites and other co-variates. Several other approaches like hierarchical clustering, supervised learning would be better to infer the findings.

4)     Discussion should be written after reanalysis and authors should try to include findings to explain disease pathogenesis based on findings to the possible extent.

This paper need to improved. However, I also like to underscore that the content of the manuscript is highly relevant and carries plethora of information on metabolic disorder pathogenesis.

Author Response

Point 1: All the sections in the manuscript should be rewritten. Introduction lacks sufficient review. I would suggest authors to review on metabolites and their implication in disease pathogenesis. The information presented in “Introduction” section is general.

Response 1: Revised in paper content in “4. Discussion” section.

Point 2: Material and method section lacks sufficient information. The preprocessing of NMR data should be explained in detail with all the parameter settings. The important point I want to highlight is quality control. Were the quality control data run during data acquisition with NMR and LC-MS? All the quality control strategy should be illustrated and the validation data should be presented as supplementary information. The raw data (excel or .csv) should be submitted as supplementary information. Biocrates kit has quality data and should be included in the manuscript as supplementary information. I would also suggest to show overall chromatogram of Biocrates standards (LC-MS, not FIA) as supplementary information.

Response 2: A detailed description of the methods is now given in the revised “Material and method” section. The preprocessing of NMR data is described in the section entitled“2.2. Nuclear Magnetic Resonance (NMR) Analysis of Plasma”. The QC data of NMR analysis are described in section entitled “2.2. Nuclear Magnetic Resonance (NMR) Analysis of Plasma” and in supplementary table 2. The QC data of MS are described in section entitled “2.3. Determining Concentrations of Plasma Metabolites with Ultra-High-Performance Liquid Chromatography-Tandem Mass Spectrometry (UPLC/MSMS)” and in supplementary figure 3. The raw data of LC-MS are now tabulated in the supplementary table 3.

Point 3: I would suggest use of more other statistical tool for analysis and infer crucial result. I also would suggest authors to use visual presentation of result than large table. For example, the authors should perform supervised and unsupervised learning to study their data. Principal component analysis would be the best way to see the data clustering. Authors have mentioned about one-way ANOVA but have not highlighted its finding in result at least as visual or tabular form. The correlation analysis may visually be presented to see the relationship between metabolites and other co-variates. Several other approaches like hierarchical clustering, supervised learning would be better to infer the findings.

Response 3: We re-analyzed all data using methods described in the section entitled “2.4. Statistical Analysis” section and each result. The unsupervised method (PCA) and supervised method (OPLS-DA) were applied to examine differences between various groups and to visualize them (figure 2A, 2B, 3A, and 3B). Additionally, weapplied the one-way ANOVA for analysis of data in table 2. Multivariable generalized linear modeling analysis was used to adjust for age and comorbidities, including hypertension, coronary artery disease (CAD), stroke and chronic kidney disease (CKD), in both NMR and MS datasets. A multiple logistic regression model with backward elimination was applied to analyse the dataset. Confounders like age, hypertension, CAD, stroke, CKD, and lipid disorder were included in the model. The statistical methods are now described in the section entitled “2.4. Statistical Analysis”.

Point 4: Discussion should be written after reanalysis and authors should try to include findings to explain disease pathogenesis based on findings to the possible extent.

Response 4: English usage has been revised.

Round  2

Reviewer 1 Report

The authors have adequately addressed the criticisms from the first review cycle, except that they appeared to misunderstood the word "medication". Medication refers to the pharmaceutical drugs the participants took. Because the drugs can greatly alter the metabolite profile, the information should be added to the manuscript if available.

Additional request is to deposit the complete dataset to a public repository upon publication, such as Metabolomics Workbench or MetaboLights.

Author Response

Dear Reviewer,

Thank you very much for your letter, indicating that our manuscript entitled “Metabolic Signature Differentiated Diabetes Mellitus from Lipid Disorder in Elderly Taiwanese” needs minor revision. 

Point 1: The authors have adequately addressed the criticisms from the first review cycle, except that they appeared to misunderstand the word "medication". Medication refers to the pharmaceutical drugs the participants took. Because the drugs can greatly alter the metabolite profile, the information should be added to the manuscript if available. 

Response 1: The participants enrolled in the present study were recruited from the individuals who resided in Chang Gung Health and Culture Village, a continuing care retirement community, for at least one year. Dietary control and physical and mental exercises have been promoted for control of chronic disease within this community. The participants of this study took questionnaire about their health, and voluntarily disclosed their medication information when they were first enrolled. After conclusion of questionnaire survey, the medication information of participants was largely incomplete, and not amenable to analysis.

Point 2: Additional request is to deposit the complete dataset to a public repository upon publication, such as Metabolomics Workbench or MetaboLights.

Response 2: Thanks for reviewer’s suggestion. The researchers who are interested in our results can contact with corresponding author.

Reviewer 2 Report

Dear authors,

Thank you for addressing the previously made comments on the manuscript "Metabolic Signature Differentiated Diabetes Mellitus from Lipid Disorder in Elderly Taiwanese". Almost all the comments have been clearly addressed, however, I would like to make suggestion for minor corrections and edits.

1) Please mention the process of data normalization for NMR and LC-MS/MS, somewhere in "Statistical Analysis" section. How were missing data handled? Please mention if any.

2) I would recommend to use two way hierarchical analysis (heat map) to give an idea about metabolome changes among different group. This would be good presentation the the result. Table 2 can be submitted as supplementary information and show top 10 significantly different metabolites as visual presentation (like boxplot) or the dot plot of the significantly different result can be presented. This way would be the best.

3) Supplementary Table S3 is not necessary to be presented in this form. Please find the visual way to present some significant result from LC-MS/MS study. Supplementary Table S3 may not be submitted. However I would suggest for deposition raw data in some of the publicly accessible database. This way it would be helpful for scientific community (This is just a suggestion).

Overall, the manuscript is well written and reveals the importance of the findings.

Thank you coming up with the good work and hope the findings for this work will be important for everyone.

Author Response

Dear Reviewer,

Thank you very much for your letter, indicating that our manuscript entitled “Metabolic Signature Differentiated Diabetes Mellitus from Lipid Disorder in Elderly Taiwanese” needs minor revision. 

Point 1: Please mention the process of data normalization for NMR and LC-MS/MS, somewhere in "Statistical Analysis" section. How were missing data handled? Please mention if any. 

Response 1: The process of data normalization for NMR is now described in section entitled “2.2. Nuclear Magnetic Resonance (NMR) Analysis of Plasma” (line 157-158).

The metabolites levels from LC-MS/MS was quantified with a linear calibration curve, which was obtained by spiking isotopically-labelled standards in plasma. The values were calculated and are expressed as μM. We didn’t normalize the quantified data from LC-MS/MS. However, the criterion we used to exclude missing data is those below the limit of detection (<LOD) or below the lower limit of quantification (<LLOQ). We have added the description to the section entitled “2.3. Determining Concentrations of Plasma Metabolites with Ultra-High-Performance Liquid Chromatography-Tandem Mass Spectrometry (UPLC/MSMS)” (line 189-191).

Point 2: I would recommend to use two way hierarchical analysis (heat map) to give an idea about metabolome changes among different group. This would be good presentation the the result. Table 2 can be submitted as supplementary information and show top 10 significantly different metabolites as visual presentation (like boxplot) or the dot plot of the significantly different result can be presented. This way would be the best.

Response 2: The heat map is presented at Figure 4 (line 275-279). Table 2 is submitted as supplementary table 3 (line 559-573). The top 10 significantly differentially abundant metabolites was presented as boxplot at figure 5 (line 282-285).

Point 3: Supplementary Table S3 is not necessary to be presented in this form. Please find the visual way to present some significant result from LC-MS/MS study. Supplementary Table S3 may not be submitted.

Response 3: Supplementary Table S3 has been replaced by the original Table 2.

Point 4: However I would suggest for deposition raw data in some of the publicly accessible database. This way it would be helpful for scientific community (This is just a suggestion)

Response 4: Thanks for reviewer’s suggestion. The researchers who are interested in our results can contact with corresponding author for downloading the dataset.